# Determining of Ablation Zone in Ex Vivo Bovine Liver Using Time-Shift Measurements

**DOI:** 10.3390/cancers15215230

**Published:** 2023-10-31

**Authors:** Mohamed Lamhamdi, Ali Esmaeili, Kiyan Layes, Zakaria El Maaroufi, Georg Rose, Andreas Brensing, Bernd Schweizer

**Affiliations:** 1Deparment of Engineering, RheinMain University of Applied Science, 65428 Rüsselsheim, Germany; ali.esmaeili@hs-rm.de (A.E.); andreas.brensing@hs-rm.de (A.B.); bernd.schweizer@hs-rm.de (B.S.); 2Institute of Medical Engineering and Research Campus STIMULATE, Otto Von Guericke University, 39106 Magdeburg, Germany; georg.rose@ovgu.de

**Keywords:** microwave ablation, ablation zone, dielectric properties, transmission measurement

## Abstract

**Simple Summary:**

Liver cancer is a very well-known disease that causes an increased death rate every year. The death rate has actually doubled in recent years. In order to increase the survival rate, it is therefore most important to detect and treat the tumor at an early stage. Microwave liver ablation (MWA) is increasingly used for successful treatment. This procedure offers the possibility of minimally invasive intervention in the treatment, so that a quick recovery time after the treatment can take place. In addition, the treatment of cancer using microwaves is a cost-effective and radiation-free treatment. This work aims to introduce an alternative method for monitoring ablation zones during thermal ablation, rather than relying on computer tomography (CT)/magnetic resonance imaging (MRI). Our results demonstrate that the measurement method provides reproducible outcomes, enabling the determination of the extent of the ablation zone.

**Abstract:**

This study presents a measurement principle for determining the size of the ablation zone in MWA, which could ultimately form an alternative to more expensive monitoring approaches like CT. The measurement method is based on a microwave transmission measurement. A MWA is performed experimentally on ex vivo bovine liver to determine the ablation zone. This setup uses a custom slot applicator performing the MWA at an operating frequency of 2.45 GHz and a custom bowtie antenna measuring the waves transmitted from the applicator. Furthermore, a custom measurement probe is used to determine the dielectric properties. A time-shift analysis is used to determine the radial extent of the ablation zone. Several measurements are carried out with a power of 50 W for 10 min to show the reproducibility. The results show that this method can provide reproducible outcomes to determine the ablation zone with a maximum error of 4.11%.

## 1. Introduction

Liver cancer (hepatocellular carcinoma, HCC) is a malignant disease of the cells in the liver. It is also referred to as “primary” liver cancer because its origin is within the liver itself. HCC is the third most common cause of cancer-related deaths in the world. A total of 841,000 new cases of HCC are estimated to have occurred in 2018, in addition to 782,000 deaths due to HCC. Over the past two decades, incidence rates have more than doubled in the USA. There has been a doubling of annual deaths from 5112 in 1999 to 11,073 in 2016. In Germany, approximately 8790 people (6160 men, 2630 women) are diagnosed with this type of cancer annually [1,2,3,4,5,6,7,8,9].

Treatment typically involves surgical resection, although it is not possible in all cases due to factors such as tumor shape, size, location, and the patient’s condition. Consequently, these limitations have led to the development of alternative approaches and therapies in recent years to reduce the global burden of cancer. One of these treatment methods is microwave ablation (MWA). MWA involves inserting a probe (applicator) into the tumor. Through this probe, microwaves are emitted at a specific power level and for a specific period of time to heat up the tumor tissue (to more than 100 °C). The duration and power of the ablation process are typically determined by the physician and the device manufacturer. Tumor volumes ranging from 3 cm to 5 cm in diameter can be effectively “ablated” by this method [10,11,12].

Microwave ablation is guided using imaging methods such as ultrasound, CT, or MRI to monitor precise applicator positioning and the spread of the ablation zone [13]. However, these imaging methods have their drawbacks as well. For instance, the use of CT can increase the risk of radiation exposure, and MRI is quite expensive and not available in all clinics [14,15].

Therefore, our research group is engaged in the development of a monitoring system for microwave liver ablation. Figure 1 illustrates a potential monitoring system. Transmitted signals from the applicator are measured via antennas positioned on the patient’s body surface. The time shift between the transmitted signal from the high-power source and the received signal from the antennas is then measured to determine the extent of the ablation zone.

Therefore, in a previous research project [16], a measurement setup was introduced to measure the time shift between the transmitted signal by the applicator at 2.45 GHz and the received signal by the antenna. Here, a measurement method was presented that determines the ablation zone. The measurements were carried out with low power on a phantom body.

To better understand microwave ablation of the liver and to develop a monitoring system to determine the ablation zone during the ablation process using time-shift measurements, this work involves ablation experiments on ex vivo bovine liver using 50 W power at a frequency of 2.45 GHz. A similar work is presented in [17], but is not directly applicable to standard microwave applicators.

## 2. Theoretical Model

### 2.1. Determining of Relative Permittivity Using an Open-Ended Coaxial Probe

In this section, the fundamentals of determining the relative permittivity of tissue using a measurement probe are explained [18,19,20]. The used measurement probe in this experiment is a self-designed probe based on the open-ended coaxial principle [21]. A calibration approach is implemented where the relative permittivity of the surrounding material at the probe tip is determined by a reflection measurement at the probe input. To calculate the relative permittivity, Equation (1) is utilized. Here$,S_{11}^{\prime }$ stands for the real part and $S_{11}^{\prime \prime }$ stands for the imaginary part of the reflection measurement. The determination of the relative permittivity $\varepsilon_{r}^{\prime }$ is performed as follows: The magnitude $S_{11}[dB]$ and phase φ of the reflection coefficient are measured with the probe and inserted into Equations (2) and (3) to calculate the real and imaginary part of $S_{11}$. These values are finally entered into Equation (1). In this context, $Z_{0}$ indicates the wave impedance of the employed coaxial cable, which is 50 Ω. The operational frequency at which the measurement probe was developed is 2.45 GHz. Here, the factor K represents the geometric factor, which is determined through calibration. The methodology for determining K is conducted in a previous work [21]. This geometric factor serves as a mathematical descriptor of the spatial field propagation of the measurement probe’s electric field. The derived value of K amounts to 3.1 × $10^{-14}$.
(1)$$\varepsilon_{r}^{\prime }=\frac{-2S_{11}^{\prime \prime }}{K{\times Z}_{0}\times \omega \times ({\left(1+S_{11}^{\prime }\right)}^{2}+{S_{11}^{\prime \prime }}^{2})}$$
(2)$$S_{11}^{\prime }=10^{\frac{S_{11}[dB]}{20}}\times cos(\varphi )$$
(3)$$S_{11}^{\prime \prime }=10^{\frac{S_{11}[dB]}{20}}\times sin(\varphi )$$

### 2.2. Determining of Ablation Zone Using TDOA Method

Figure 2 illustrates a theoretical model that can be used to calculate the linear expansion of a heated material zone during MWA. This model abstracts from the real ablation situation by assuming only two tissues states: cold tissue and completely ablated tissue. It includes scenarios both before (T = 0) and during (T > 0) ablation [22]. In this context, cold tissue represents the tissue before ablation with a permittivity of $\overset{\prime }{\varepsilon_{r\_cold}}$ and hot tissue represents the tissue after ablation with a permittivity of $\overset{\prime }{\varepsilon_{r\_hot}}$. As shown in [23], the propagation of the ablation zone is determined by a mathematical model. To determine the ablation zone, the knowledge of the relative permittivity of the tissue before ablation and the relative permittivity of the ablated tissue is required. Microwave ablation causes dehydration of the tissue and therefore an increase in tissue temperature. As the temperature of the tissue changes, the permittivity of the tissue also changes. This affects the propagation velocity in the tissue. Using the TDOA method, Equations (1)–(4) from [23] can be used to calculate the extent of the ablation zone.

The radial distance, denoted as $r_{AB}$, between points A and B serves as a representation of the size of the ablation zone. This distance is mathematically deduced from the measured time shifts and subsequent arrival time disparities of the transmitted signal throughout the ablation process. Equations (4)–(6) are the key formulas from [23] that guide the calculation of $r_{AB}$.
(4)$$r_{AB}=F\times v_{0}\times ∆t$$
(5)$$v_{0}=\frac{c_{0}}{\sqrt{\varepsilon_{r\_cold}}}$$
(6)$$F=\frac{1}{1-\sqrt{\frac{\varepsilon_{r\_hot}}{\varepsilon_{r\_cold}}}}$$

In Equation (6), the parameter F indicates the tissue permittivity associated with both cold and hot tissue. Ideally, in practical applications, these values should be measured directly during the ablation. In our experimental configuration, we are able to empirically measure the permittivity of the tissue before and at the end of the ablation. The term $v_{0}$ represents the propagation velocity of microwaves within the cold tissue. The quantity ∆t denotes the change in time delay between the transmitted and received signal. A measurement setup was therefore designed to measure this time delay. This model can also be applied to tumour tissue, as the changes in the propagation velocity of microwaves are due to the different permittivity compared to healthy tissue. This would be noticeable when measuring ∆t.

## 3. Experimental Method and Results

### 3.1. Experimental Setup

The ablation system used is an experimental ablation system consisting of a self-developed high-power source designed in a previous work within this research group. In addition, a self-developed single-slot applicator without cooling mechanism, based on the λ/4-transformer theory, and a self-developed bow-tie body matched antenna were also developed for this ablation system by our research group [24].

For the experimental setup of microwave ablation processes on ex vivo bovine liver, the following measurement configuration is employed (see Figure 3). This setup enables the measurement of time shift between two signals: the signal transmitted to the tissue and the signal received by the antenna. By measuring the time shift, the interaction between the transmitted microwave signal and the tissue can be analyzed. This information is crucial for assessing the effectiveness of microwave ablation and precisely controlling the ablation process.

Carrying out the ablation process on bovine liver and measuring the time shift poses two challenges. The first challenge arises from the requirement of a high power of 50 W for ablation, which could potentially be dangerous for other measurement instruments and lead to their damage. To solve this issue, a directional coupler is utilized. Here, an attenuated signal from the signal transmitted to the liver is extracted as a probe and fed back into the measurement setup as a reference signal through the coupling port of the directional coupler. This reference signal can then be used for further measurements and analysis. It is important to note, however, that due to the hazards associated with working at such high-power levels, all measurements were conducted within an anechoic chamber for safety reasons.

The second challenge arises from the high frequency of 2.45 GHz used in this measurement, as the agilent technologie DSO5054A oscilloscope (santa clara, CA, USA) used can only display signals up to a frequency of 500 MHz. To overcome this limitation, a frequency mixer ADL5802-EVALZ from analog devices (Norwood, MA, USA) is employed. In this process, the high-frequency signal is multiplied by another high-frequency signal produced over a local oscillator DSG836 from RIGOL technologies (Suzhou, people´s republic of china), and after filtering with low pass filters VLF-5850+ from mini-circuits (brooklyn, NY, USA), the resulting low-frequency signal is sent to the oscilloscope for further processing and analysis.

By combining the directional coupler and frequency mixer, both the high-power and high-frequency requirements for measuring time shift and analyzing the ablation process can be met. This enables the precise and safe execution of experiments for characterizing ablation processes on bovine liver, particularly the measurement of time shifts, contributing to the advancement of this medical treatment method. The following sections provide a more detailed explanation of this measurement setup.

In this experimental setup, a high-power signal with an amplitude of 50 W and a frequency of 2.45 GHz is fed through a panel into the input port of the directional coupler located within the anechoic chamber. The aim is to transmit this signal from the output port of the directional coupler to the applicator previously inserted into the liver. The signal is transmitted at the same power and frequency.

The coupling port of the directional coupler extracts a −40 dB part of the input signal and directs it to the mixer (RF1). There, it is multiplied with another signal having a power of 0 dB and a frequency of 2 GHz. This additional signal is generated by another signal generator and fed through the local oscillator port of the mixer. The multiplication of these two signals results in a new signal emitted from the first output of the mixer (IF1). This signal encompasses various frequency components, mainly $f_{1}$ = 2.45 GHz − 2.0 GHz and $f_{2}$ = 2.45 GHz + 2.0 GHz. To eliminate unwanted frequency components, the signal is filtered after mixing using a low-pass filter. This filter only allows lower frequencies to pass through while suppressing undesired frequency components. The filtered signal with a frequency of $f_{1}$ = 450 MHz is then directed to the first channel of the oscilloscope. This signal, serving as a probe of the transmitted signal, is utilized as a reference signal to measure time shifts during ablation.

The ablation process initiates when the transmitted signal is emitted by the applicator and interacts with the bovine liver. The signal is received through a body-matched antenna placed on the liver opposite the slot of the applicator. However, the signal is attenuated by up to 40 dB due to tissue penetration and absorption. Despite this attenuation, the signal retains its original frequency of 2.45 GHz. Consequently, the signal is forwarded through another port of the panel to the second input of the mixer (RF2). There, it is once again multiplied with the local oscillator signal to generate a lower frequency signal. The resulting signal is then output through the second output of the mixer (IF2).

To eliminate unwanted frequency components and down-convert the signal to a displayable frequency, it is also filtered after mixing with the local oscillator signal using a low-pass filter. This again produces a signal with a frequency of 450 MHz, which is directed to the second channel of the oscilloscope. This signal is regarded as the received signal and serves for measuring time shifts in comparison to the transmitted signal.

Through observing and analyzing the time shift of the received signal compared to the transmitted signal during ablation, valuable insights into tissue conditions and changes can be gained. Measuring time shifts facilitates the precise monitoring of the ablation process progression and assessment of treatment efficacy.

### 3.2. Experimental Procedure

Step 1: Cutting of Bovine Liver and Measuring of $S_{11}$ Values before Ablation

In the first step (see Figure 4a), a piece of bovine liver at the starting temperature of approximately 15 °C is cut off. Care is taken to ensure that no large vessels visible from the outside run through the section. The height and width of the tissue sample should also be greater than the expected short-axis diameter (SAD) of the ablation zone. The expected value for SAD is a maximum of 4 cm. These estimates are based on experience. Next, the reflection measurement on the vector network analyzer ZVL13 (Rohde&Schwarz, München, Germany)is prepared. First, a complete calibration of the vector network analyzer to the end of the cable is performed. To carry this out, a calibration procedure is started on the vector network analyzer and then the calibration kit is screwed onto the cable end as required. Then, the specially developed measuring probe is connected to the cable end. This measuring probe is developed and characterized according to the principle of a coaxial-based dielectric measuring probe with an open end [14]. Calibration to the end of the probe is conducted by short-circuit calibration. Once calibration is complete, repeat measurements of the reflectance parameter $S_{11}$ are performed. For this purpose, the tip of the measuring probe is held on the cut surface of the liver sample and the $S_{11}$ value is noted. This is repeated 10 times for a fixed point. The mean value is formed from the measured values of magnitude and phase and entered into the Equations (2) and (3). With Equation (1), it is possible to determine the relative permittivity $\overset{\prime }{\varepsilon_{r\_cold}}$ of the liver tissue before ablation.

Step 2: Positioning of applicator

The second step in the measurement process is the positioning of the applicator (see Figure 4b). Initially, an insertion hole for the applicator is pre-drilled with a wooden stick. Care is taken to ensure that the wooden stick has the same diameter as the applicator itself. After, that the wooden stick is removed and the applicator is inserted into the pre-drilled hole.

Step 3: Place the bovine liver sample in the absorber chamber

The third step takes place inside the anechoic chamber where the actual experiment is conducted (see Figure 4c). The liver sample with applicator is placed here. A panel connects the N-cables outside and inside the anechoic chamber. Inside the anechoic chamber, the body-matched antenna and the applicator are connected to the cables. The body-matched antenna is placed on the surface of the bovine liver centered above and parallel to the applicator’s slot. It is ensured that there is no air gap between the bovine liver surface and body-matched antenna. After closing the anechoic chamber, the voltage supply for the mixer and the frequency (2 GHz) and power level (0 dBm) of the local oscillator are set. Then, the laptop is connected to the high-power source and the parameters of the experiment are set via the GUI on the laptop. These parameters are the following: an operating frequency of 2.45 GHz and a power of 50 W. Now, the microwave ablation is started. For this purpose, both the operating frequency and the power are released at the output via the GUI. During the ablation, the time difference ∆t at the frequency 450 MHz between the peaks of the transmitted signal and the received signal is measured via the oscilloscope in a time interval of 30 s. This measured value corresponds to the time difference between the peaks of the transmitted signal and the received signal at the frequency of 450 MHz. These measured values must be converted to the frequency at 2.45 GHz. A total of 21 measurements of ∆t are recorded at the frequency of 450 MHz for an ablation time of 10 min. After recording the last measured value, the ablation is terminated by switching off the signal output of the high-power source.

Step 4: Cut open the bovine liver and measure the $S_{11}$ values and the ablation zone after ablation

The fourth step is also the last one within the measurement sequence (see Figure 4d,e). Here, the anechoic chamber is opened and the applicator and body-matched antenna are disconnected from the cable connections. Then, the bovine liver is removed from the anechoic chamber together with the applicator and placed in front of the vector network analyzer. The tissue sample is then cut open along the applicator. The applicator is then carefully removed from the bovine liver. Now the reflection factor is determined by means of ten repeated measurements in the center of the ablation zone, measured approximately 5 min after the end of the ablation. The center of the ablation zone is the place where the slot of the applicator was located. From the mean value of magnitude and phase, the relative permittivity $\overset{\prime }{\varepsilon_{r\_hot}}$ of the ablated bovine liver are calculated analogously to Step 1. Finally, the SAD are measured on both cut surfaces.

To verify the reproducibility of the measurement results, the experimental series is conducted a total of seven times, with similarly sized liver samples, under identical conditions. In the next subsection, all measurements are presented comprehensively. These measurements facilitate a detailed characterization of the ablation zone and provide valuable insights for further analysis and interpretation of the conducted ablation processes. Additionally, they offer crucial information for monitoring these processes.

### 3.3. Experimental Results

Table 1 presents the results of reflection measurements on cold bovine liver tissue. Both the magnitude and phase of the $S_{11}\;$ values are recorded. The magnitude and phase values of $S_{11}$ are each averaged from 10 repeated reflection measurements for each sample. Using these averaged $S_{11}\;$ values, the relative permittivity $\overset{\prime }{\varepsilon_{r\_cold}}$ (see Equation (1)) is calculated.

Table 2 presents the results of reflection measurements on bovine liver tissue at the center of the ablation zone. Both the magnitude and phase of the $S_{11}$ values are recorded. The magnitude and phase values of $S_{11}$ are each averaged from 10 repeated reflection measurements conducted in Step 1 (refer to Figure 4d). Using these measured values, the relative permittivity $\overset{\prime }{\varepsilon_{r\_hot}}$ (see Equation (1)) is calculated.

Figure 5 illustrates the changes in time delay ∆t between the transmitted and received waves from 7 measurements during a 10 min ablation at a frequency of 2.45 GHz (therefore, the time delay before the ablation is set to 0, and all following time delays are given relative to this first value, thus indicating the change in time delay). It is important to note that the change in time delay is measured at 450 MHz, and the measured time delay needs to be converted to the frequency range of 2.45 GHz. The changes in time delay are depicted on the *y*-axis in picoseconds (ps), while the ablation time is plotted on the *x*-axis in seconds (s). Each curve represents an individual measurement, and each point on the curves indicates the specific time-delay change at a particular moment during the ablation process. The delay values increase from zero to approximately 30 to 50 ps. In measurement 6, in the first 150 s after the ablation process, rises and falls occur in the curves. A possible cause for these rises and falls could be that there are veins or air bubbles between the applicator slot and the antenna in the path of the electromagnetic waves, which were not previously detectable by visual observation.

Table 3 presents the values of the measured radius, the change in time delay between the transmitted and received signals, and the calculated radius of the ablation zone. The calculated radius in column 4 is determined using Equation (4). In this calculation, the respective permittivity of the cold and hot tissue was taken from Table 1 and Table 2. The measured radius is half of the SAD. To evaluate the deviations between the measured and calculated radius, the relative error is provided, expressed as a percentage.

As shown in Table 3, the error ranges from 65.35% to 75%, with a mean value of 70.23% and a standard deviation of 4.11%. The low standard deviation of the error suggests that the discrepancies between the measured and calculated radius are primarily attributed to the theoretical model (TDOA method) or the calculation formula. This suggests the possibility of compensating for the average error of 70.23% by introducing a correction factor into Equation (4). The deviation of 70.23% corresponds to a correction factor $K_{correction}=\frac{r_{measured}}{r_{calculated}}$.

To observe the effect of the correction factor on independent measurements, this correction factor is determined in Table 4 over the first four measurements and applied to measurements 5, 6, and 7. The correction factor is $K_{correction}=\frac{14.81\;mm}{4.52\;mm}=3.28$. After applying the correction factor to measurement 5, 6, and 7, the mean value of relative error is 11.9% and the maximum value is 18%. This results in a maximum deviation of 2.7 mm in the radius of the ablation zone.

## 4. Discussion

The following section presents the obtained results for discussion. In this work, 7 MWAs were performed in bovine liver with a power of 50 W and an ablation time of 10 min to show the reproducibility of the measurement. Before ablation, the permittivity of the tissue $\overset{\prime }{\varepsilon_{r\_cold}}$ was determined using a reflectance probe. During ablation, the time delay ∆t between the transmitted wave and the received wave was measured. At the end of the ablation, the permittivity $\overset{\prime }{\varepsilon_{r\_hot}}$ was determined using the reflection probe. Using the permittivity before ablation $\overset{\prime }{\varepsilon_{r\_cold}}$, the permittivity after ablation $\overset{\prime }{\varepsilon_{r\_hot}}$, and the change in time delay ∆t, the extent of the ablation zone was determined using Equation (4). This calculated zone radius was compared with the actual zone measured by cutting open the liver. This discussion focuses on the change in the time ∆t, the determination of the permittivity before ablation $\overset{\prime }{\varepsilon_{r\_cold}}$, the determination of the permittivity after ablation $\overset{\prime }{\varepsilon_{r\_hot}}$, and the average error of 70.23 % between the measured radius and the calculated radius of the ablation zone across all measurements (see Table 3). In addition, measures for translating this approach to the clinics are discussed.

The measurements showed that the permittivity before ablation $\overset{\prime }{\varepsilon_{r\_cold}}$ has a mean value of 40.29 with a standard deviation of 1.02. This small standard deviation shows a reproducibility of the measurements. The measured values of $\overset{\prime }{\varepsilon_{r\_hot}}$ show a mean of 15.36 and a standard deviation of 4.42. It can be noted that the standard deviation in this case is significantly larger than before the ablation. This larger variation contributes to an error in the calculation of the ablation zone.

One other potential source of error that could lead to discrepancies between the measured and calculated radius is the bad coupling of the applicator to the liver tissue during an ablation. For instance, if an air bubble forms during ablation [25,26,27], it can change the layers of the transmission medium that the transmitted signal must traverse. In such a scenario, erroneous ∆t measurements may result. One potential way to detect this source of error could be through conducting reflection measurements at the applicator during the ablation process.

An additional potential source of error could be the erroneous reading of ∆t on oscilloscope (because of the high sensibility of the measurement setup and operation at high frequencies). In this case, an automated measurement of ∆t, such as through a software-defined radio system (SDR), could help reduce the error [28,29].

Furthermore, the measurements revealed that the relative permittivity within the ablation zone is not homogeneous. Specifically, it was observed that the relative permittivity at the center of the ablation zone (at the slot of the applicator) has the lowest value, while it increases with increasing distance from the center and approaching the edge of the ablation zone (see Figure 6). This behavior is logical and expected. However, the theoretical model assumes that the relative permittivity is constant throughout the ablation zone. It also assumes a distinct and clearly visible boundary between cold and hot tissue, which is not observed in reality.

This layered variation in relative permittivity can lead to deviations between the measured and calculated radius. These aspects are not adequately considered in the theoretical model. A possible solution could be a more detailed examination of the permittivity distribution within the ablation zone. For instance, an exemplary calculation of all measurements using the average permittivity of $\overset{\prime }{\varepsilon_{r\_hot\_new}}$ (calculating the average between the permittivity of cold tissue $\overset{\prime }{\varepsilon_{r\_cold}}$ and the permittivity of hot tissue $\overset{\prime }{\varepsilon_{r\_hot}}$ at the slot) is carried out (refer to Table 5). As observed from the Table 4, a reduction of approximately 40% in the mean error was achieved through the modified model considering the altered permittivity distribution of hot tissue $\overset{\prime }{\varepsilon_{r\_hot\_new}}$.

For a more accurate determination of the radius of the ablation zone, measuring the relative permittivity immediately after the completion of ablation is also necessary [30,31]. However, this measurement is not feasible within the current measurement procedure. It requires the liver to be removed, dissected, and then the permittivity measurement carried out. During this time frame, there could be a buildup of water in the tissue, which in turn could alter the dielectric properties and relative permittivity of the ablation zone. One possible solution to mitigate this source of error could be conducting permittivity measurements during the ablation process.

The tissue shrinkage that occurs during ablation also leads to an incorrect prediction of the ablation zone [32,33]. This model would have to be modified to take this aspect into account. A possible modification of this system could be the support by imaging systems or the addition of an empirical shrinkage model.

What must also be taken into account here is that the theoretical model (TDOA method) assumes a spherical expansion of the heat front and the applicator used in this experiment causes a drop-shaped expansion of the ablation zones. This also causes measurement inaccuracies which lead to a deviation between the measured radius and the calculated radius. A possible improvement of the system could be the development of a new applicator with possible cooling.

In order to translate this measurement method to a clinical situation, a number of adaptations are required; however, the underlying signal mechanism is generally suitable for use in real patients. For this purpose, a body-matched antenna suitable for signal detection on the human skin would have to be developed. With the antenna used in this study, it would be necessary to use a coupling medium between the antenna and the skin. Furthermore, the antenna would have to be sensitive enough to tolerate the attenuation losses that the transmitted wave experiences during its passage from the applicator to the patient surface. Further on, a more precise evaluation of the measurement data could be carried out by extending the measurement setup to a multiple antenna array located at the patient surface. This would allow for redundancy of the ∆t data and a more accurate prediction of the extent of the ablation zone in different directions. In addition, extensions of the method to generate a permittivity profile within the entire ablation zone during the MWA are currently under investigation. This would offer the advantage that real-time monitoring of the ablation zone could be carried out with smaller equipment requirements than today.

## 5. Conclusions

In this paper, a measurement method is presented that determines the propagation of the ablation zone which occurs during MWA. The method is tested in practice with ex vivo bovine liver. The measured time delays show a strongly reproducible correlation with the ablation zone radii. Hence, it can be concluded that the time-shift method can be used for monitoring of MWA. The theoretical model currently gives smaller hot zone radii by a systematic factor of about 3. Further studies will be conducted to resolve this deviation between theory and measurement, including approaches like measuring the reflection at the location of the applicator itself, in situ measuring of the permittivity at the ablation zone during an ablation, as well as employing modified theoretical models.

## Figures and Tables

**Figure 1 cancers-15-05230-f001:**
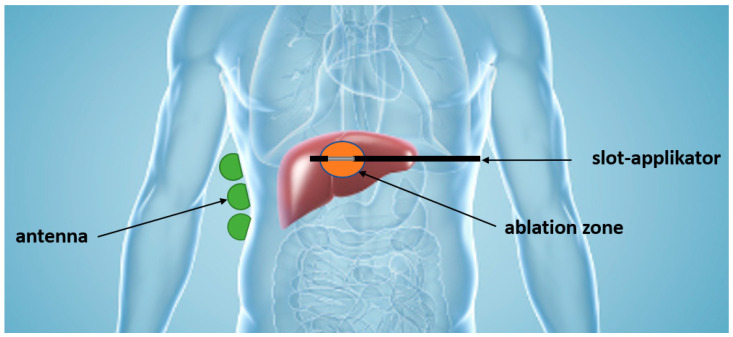
Schematic drawing of the measurement setup which can be used to monitor the ablation in MWA.

**Figure 2 cancers-15-05230-f002:**
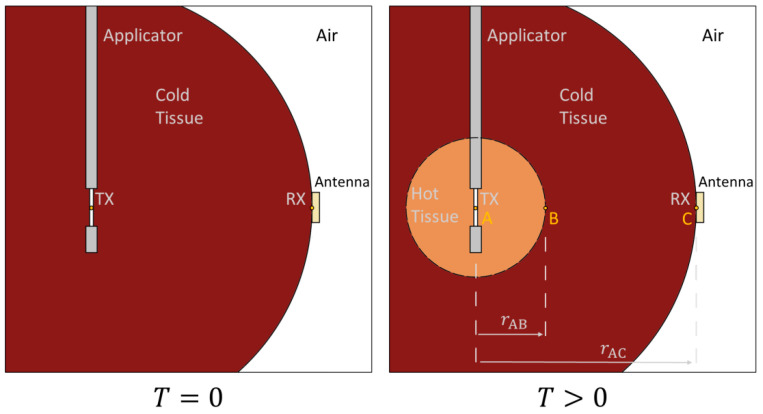
Block diagram of the test setup for measuring the ablation zone over the time delay between the emitted and received electromagnetic wave.

**Figure 3 cancers-15-05230-f003:**
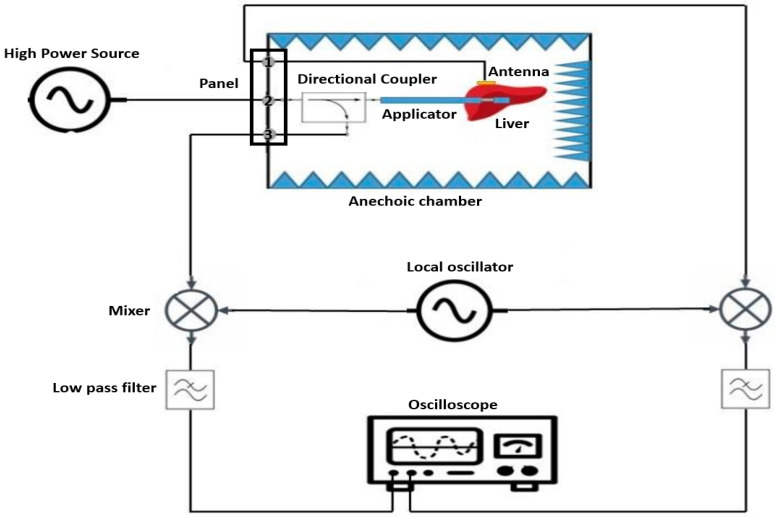
Schematic design of time-shift measurement setup during a MWA.

**Figure 4 cancers-15-05230-f004:**
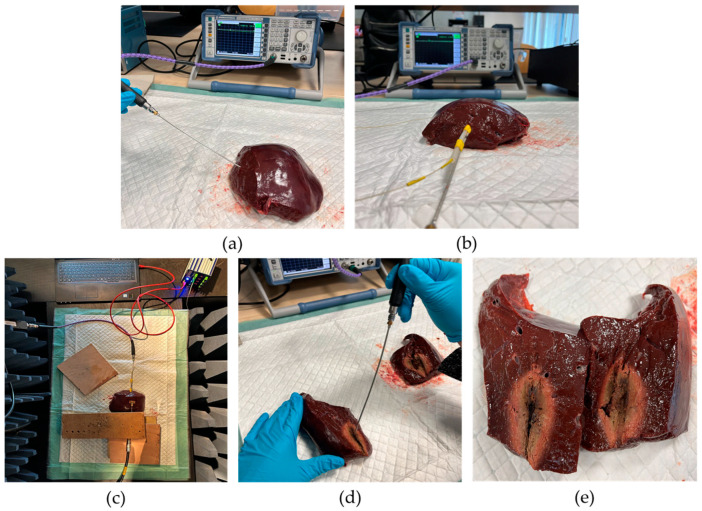
The overview of each station: (**a**) Step 1; (**b**) Step 2; (**c**) Step 3; (**d**,**e**) Step 4.

**Figure 5 cancers-15-05230-f005:**
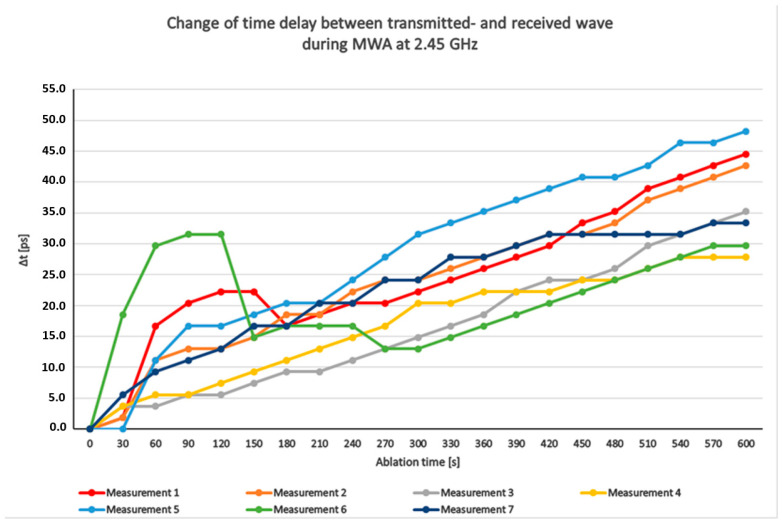
Measured changes of time delay between transmitted- and received wave during a MWA.

**Figure 6 cancers-15-05230-f006:**
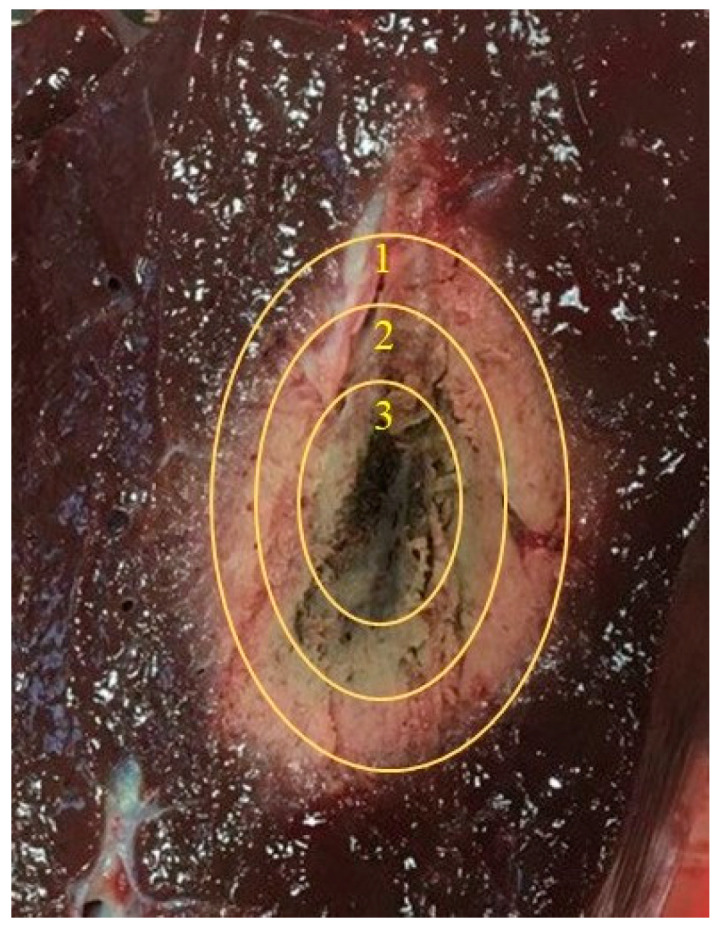
Representation of the permittivity distribution in different regions of the ablation zone: (1) $\overset{\prime }{\varepsilon_{r}}$ = 30; (2) $\overset{\prime }{\varepsilon_{r}}$ = 23.2; (3) $\overset{\prime }{\varepsilon_{r}}$ = 12.4.

**Table 1 cancers-15-05230-t001:** Measurement results for $S_{11}$ parameters and the relative permittivity of cold tissue.

Measurement	$S_{11}$		$\overset{\prime }{\varepsilon_{r\_cold}}$
	Magnitude [dB]	Phase [°]	
1	−3.43	−94.4	41.71
2	−3.28	−92.3	40.67
3	−3.18	−92.9	41.14
4	−3.20	−91.0	39.99
5	−2.60	−88.2	38.91
6	−2.46	−88.3	39.14
7	−2.44	−90.3	40.51
Mean value	−2.94	−91.05	40.29
Standard deviation	0.42	2.32	1.02

**Table 2 cancers-15-05230-t002:** Measurement results for $S_{11}$ parameters and the relative permittivity of hot tissue.

Measurement	$S_{11}$		$\overset{\prime }{\varepsilon_{r\_hot}}$
	Magnitude [dB]	Phase [°]	
1	−2.62	−35.3	12.97
2	−2.47	−40.2	15.01
3	−2.92	−52.5	19.96
4	−2.41	−43.4	16.31
5	−0.82	−24.6	9.13
6	−2.13	−55.9	21.81
7	−1.45	−33.1	12.37
Mean value	−2.11	−40.71	15.36
Standard deviation	0.73	10.98	4.42

**Table 3 cancers-15-05230-t003:** Comparison of the experimental results with the results of the theoretical model.

Measurement	∆t [ps]	Measured Radius [mm]	Calculated Radius [mm]	Relative Error [%]
1	44.5	14.5	4.66	67.86
2	42.7	14.75	5.11	65.35
3	35.2	16	5.41	66.18
4	27.8	14	3.64	74
5	48.2	17.5	4.49	74.34
6	29.6	18	5.59	68.94
7	33.3	14	3.5	75
Mean value	37.3	15.53	4.63	70.23
Standard deviation	7.85	1.66	0.82	4.11

**Table 4 cancers-15-05230-t004:** Comparison of the experimental results with the modified results of the theoretical model.

Measurement	∆t [ps]	Measured Radius [mm]	Calculated Radius [mm]	Relative Error [%]
1	44.5	14.5	4.49	67.86
2	42.7	14.75	5.59	65.35
3	35.2	16	3.5	66.18
4	27.8	14	4.49	74
Mean value	37.55	14.81	4.52	68.35
Standard deviation	7.65	0.85	0.85	3.91
**Measurement**	∆t **[ps]**	**Measured Radius [mm]**	**New Calculated** **Radius [mm]**	**Relative Error** **[%]**
5	48.2	17.5	14.72	15.9
6	29.6	18	18.34	1.9
7	33.3	14	11.48	18
Mean value	37.0	16.5	14.85	11.9
Standard deviation	9.84	2.18	3.43	8.75

**Table 5 cancers-15-05230-t005:** Comparison of the modified results.

Measurement	$\overset{\prime }{\varepsilon_{r\_cold}}$	$\overset{\prime }{\varepsilon_{r\_hot\_new}}$	∆t [ps]	Measured Radius [mm]	New Calculated Radius [mm]	Relative Error [%]
1	41.71	27.34	44.5	14.5	10.85	25
2	40.67	27.84	42.7	14.75	11.62	21
3	41.14	30.54	35.2	16	11.88	25
4	39.99	28.15	27.8	14	8.11	41
5	38.91	24.02	48.2	17.5	10.95	37
6	39.14	30.47	29.6	18	12.05	30
7	40.5	26.43	33.3	14	8.16	41
Mean value	40.28	27.82	37.3	15.53	10.52	31.42
Standard deviation	1.01	2.27	7.85	1.66	1.67	8.24

## Data Availability

The data presented in this study are available in this article.

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
