# Peer review of "Determining of Ablation Zone in Ex Vivo Bovine Liver Using Time-Shift Measurements"

_cancers, 2023, doi:10.3390/cancers15215230_

Round 1

Reviewer 1 Report

Comments and Suggestions for Authors

This paper presents a methods for estimating the ablation zone during microwave ablation. Such a method could have clinical benefit, if it can be translated into human patients. There are a number of issues that should be addressed, including some discussion on what would be necessary to translate the proposed system into humans and what problems there may arise. In addition, some sections could benefit from revision as described in more detail below.

Detailed comments:

-ln 36: add worldwide, and maybe US and EU numbers as well in addition to Germany

-change 'burnt' to 'ablated' - 'burnt' is not commonly used in the ablation literature

-ln 65-89: shorten, or remove. A paper summary in the introduction is not typically included, as shown here.

-Equ 1,2,3: clarify in text, what variables are measured and which are calculated.

-dielectric properties has a substantial temperature dependence, and not only depend on whether ablated or not (there may also be a dependence on temperature history, e.g., Arrhenius type relationship). In any case, this needs to be discussed since the temperature varies throughout the ablation zone, as well as surrounding the ablation. epsilon_r_cold and _hot are therefore not constant as assumed.

-please describe in more detail what an 'anechoic chamber' is, and why this was required.

-there aren't any details provided on the MW applicator. Was this a clinical MW ablation system, or an experimental one? What kind of antenna was used? Was it cooled?

-Please reduce abbreviations throughout to those that are frequently used (e.g., HPS, OS, LO, etc could be removed since it's difficult to remember all those abbreviations when reading the manuscript.

-what was the initial temperature of the liver?

-Table 1,2: please add mean and stddev

-at what location was er_hot measured exactly? what was the temperature of the liver, specifically the measurement location at that time?

-Fig. 5: why is delta_t=0 at t=0? Based on the descriptions, even at the beginning there should be a time difference present.

-one limitation is that only the final ablation diameter is validated, and it is not clear whether predictions of ablation zone size at earlier times are accurate.

-the discussion section needs a major revision. There is no introduction, and the discussion focuses on the large error. There should be substantial discussion on additional results, and particularly on the clinical utility: what is necessary to translate this method into clinics - does the system need to be modified? will the method be accurate if measurements are taken on the skin surface instead on the liver? will the signal be sufficiently clean? Could you use multiple antennas to obtain more data or get more accurate predictions? How would this method help in the clinic?

-please provide more details on the correction factor noted below Table 3. How was the factor calculated? Can you add corrected results in Table 3? Is this correction different from what you present in Table 4?

-dielectric properties of tumor tissue may be different from normal liver - cirrhosis may also change properties. how will this affect the accuracy of your method?

-in addition, some method sections appear lengthy and can be reduced in length

-the following paper presents a related method for monitoring MW ablations, and should be cited:
'Electromagnetic Transmission Coefficient-Based Assessment of Tissue State During Microwave Ablation', Frontiers in Biomedical Devices, 2022

Comments on the Quality of English Language

Quality of english language is ok.

Author Response

Dear Mr Reviewer 1,
first of all I would like to thank you for your constructive comments. Your comments lead to an upgrading of the quality of the paper. Enclosed you will find the revised version. I have marked the changed parts of your comments with red text. I have also indicated the line numbers below. 

-ln 36: add worldwide, and maybe US and EU numbers as well in addition to Germany

line 34 - 41

-change 'burnt' to 'ablated' - 'burnt' is not commonly used in the ablation literature

line 51

-ln 65-89: shorten, or remove. A paper summary in the introduction is not typically included, as shown here.

line 66-69

-Equ 1,2,3: clarify in text, what variables are measured and which are calculated.

line 78-94

-dielectric properties has a substantial temperature dependence, and not only depend on whether ablated or not (there may also be a dependence on temperature history, e.g., Arrhenius type relationship). In any case, this needs to be discussed since the temperature varies throughout the ablation zone, as well as surrounding the ablation. epsilon_r_cold and _hot are therefore not constant as assumed.

line 96-108

-please describe in more detail what an 'anechoic chamber' is, and why this was required.

line 148-150

-there aren't any details provided on the MW applicator. Was this a clinical MW ablation system, or an experimental one? What kind of antenna was used? Was it cooled?

line 129-133

-Please reduce abbreviations throughout to those that are frequently used (e.g., HPS, OS, LO, etc could be removed since it's difficult to remember all those abbreviations when reading the manuscript.

line 155-270

-what was the initial temperature of the liver?

line 201-202

-Table 1,2: please add mean and stddev

line 276-284

-at what location was er_hot measured exactly? what was the temperature of the liver, specifically the measurement location at that time?

line 253- 256

-Fig. 5: why is delta_t=0 at t=0? Based on the descriptions, even at the beginning there should be a time difference present.

line 285-288

-one limitation is that only the final ablation diameter is validated, and it is not clear whether predictions of ablation zone size at earlier times are accurate.

-the discussion section needs a major revision. There is no introduction, and the discussion focuses on the large error. There should be substantial discussion on additional results, and particularly on the clinical utility: what is necessary to translate this method into clinics - does the system need to be modified? will the method be accurate if measurements are taken on the skin surface instead on the liver? will the signal be sufficiently clean? Could you use multiple antennas to obtain more data or get more accurate predictions? How would this method help in the clinic?

line 324-409

-please provide more details on the correction factor noted below Table 3. How was the factor calculated? Can you add corrected results in Table 3? Is this correction different from what you present in Table 4?

line 314-323

-dielectric properties of tumor tissue may be different from normal liver - cirrhosis may also change properties. how will this affect the accuracy of your method?

line 123-126

-in addition, some method sections appear lengthy and can be reduced in length

-the following paper presents a related method for monitoring MW ablations, and should be cited:
'Electromagnetic Transmission Coefficient-Based Assessment of Tissue State During Microwave Ablation', Frontiers in Biomedical Devices, 2022

line 74-75

Reviewer 2 Report

Comments and Suggestions for Authors

In this manuscript, the authors introduce a method for monitoring ablation zones during thermal ablation to determining the size of the ablation zone in MWA. The topic of this manuscript is meaningful. I recommend this work to be accepted after the following questions being addressed:

[1] Does thermal expansion, followed by tissue shrinkage after ablation, introduce any potential bias into the measurements? To ensure the accuracy of ablation predictions and assessments, what strategies can be employed to mitigate the influence of tissue shrinkage on measurements?

[2] Will the non-uniform, teardrop-shaped radiation pattern in the ablation zone of bovine liver affects measurement inaccuracies? Existing techniques cannot control the length, width or Angle of ablation. How does the TDOA method assess the ablation area's shape and mitigate measurement errors arising from diverse shapes?

[3] In Figure 5, why did measurement 6 rise and fall sharply in the first 150 seconds of ablation compared with other groups?

[4] It is suggested to supplement the concrete structure diagram of the measuring device.

Author Response

Dear Mr Reviewer 2,
first of all I would like to thank you for your constructive comments. Your comments lead to an upgrading of the quality of the paper. Enclosed you will find the revised version. I have marked the changed parts of your comments with blue text. I have also indicated the line numbers below.    

[1] Does thermal expansion, followed by tissue shrinkage after ablation, introduce any potential bias into the measurements? To ensure the accuracy of ablation predictions and assessments, what strategies can be employed to mitigate the influence of tissue shrinkage on measurements?

line 385-388

[2] Will the non-uniform, teardrop-shaped radiation pattern in the ablation zone of bovine liver affects measurement inaccuracies? Existing techniques cannot control the length, width or Angle of ablation. How does the TDOA method assess the ablation area's shape and mitigate measurement errors arising from diverse shapes?

line 389-394 and line 402-405

[3] In Figure 5, why did measurement 6 rise and fall sharply in the first 150 seconds of ablation compared with other groups?

line 295-298

[4] It is suggested to supplement the concrete structure diagram of the measuring device.

I did not understand this proposed amendment. Can you explain this comment in more detail?

best regard 

Mohamed Lamhamdi
